# Implementation of a Sensor Big Data Processing System for Autonomous Vehicles in the C-ITS Environment

**Aelee Yoo**, **Sooyeon Shin, Junwon Lee and Changjoo Moon** *

Department of Smart Vehicle Engineering, Intelligent Data Processing laboratory, Konkuk University,
120 Neungdong-ro, Gwangjin-gu, Seoul 05029, Korea; aelee1010@konkuk.ac.kr (A.Y.);
dge05207@konkuk.ac.kr (S.S.); jileejun@konkuk.ac.kr (J.L.)
* Correspondence: cjmoon@konkuk.ac.kr

**Abstract:** To provide a service that guarantees driver comfort and safety, a platform utilizing connected car big data is required. This study first aims to design and develop such a platform to improve the function of providing vehicle and road condition information of the previously defined central Local Dynamic Map (LDM). Our platform extends the range of connected car big data collection from OBU (On Board Unit) and CAN to camera, LiDAR, and GPS sensors. By using data of vehicles being driven, the range of roads available for analysis can be expanded, and the road condition determination method can be diversified. Herein, the system was designed and implemented based on the Hadoop ecosystem, i.e., Hadoop, Spark, and Kafka, to collect and store connected car big data. We propose a direction of the cooperative intelligent transport system (C-ITS) development by showing a plan to utilize the platform in the C-ITS environment.

**Keywords:** autonomous vehicle; LDM; Hadoop ecosystem; connected car; big data platform; IoT; C-ITS

## 1. Introduction

Internet of Things (IoT) is a technology in which all its elements, such as humans, objects, and services, are connected through wireless communication, and an object capable of wireless communication with a sensor is referred to as an IoT edge device. The core characteristic of IoT is connectivity. New services are provided to users by collecting and sharing new information generated by IoT edge devices between devices. Therefore, IoT is one of the significant foundation technologies of the 4th industrial revolution, where various types of convergence and exchange through innovation are important.

IoT technology has gradually evolved, and its application range has expanded from devices used in everyday life, such as smartwatches, to connected cars. A connected car uses wireless communication technology and exchanges information with connected cars and infrastructure. Despite the progress and development of research on autonomous driving technology that autonomously recognizes and determines the surrounding environment through sensors, such as radar, LiDAR, GPS, and camera, autonomous vehicles still do not guarantee safety [1]. If connected technology is applied to an autonomous vehicle, it is possible to provide a service that guarantees driver comfort and safety. Therefore, in this study, an autonomous vehicle with connected technology is set as an IoT edge device. In order to provide IoT services to IoT edge devices, a platform that can collect and process all data generated by autonomous vehicles is needed.

In the automotive field, the environment in which autonomous vehicles are connected to other vehicles and infrastructure through wireless communication is defined as a cooperative intelligent

transport system (C-ITS). C-ITS changed the existing ITS from road management to user safety through smooth two-way communication and collaboration between vehicles, roads, and improved safety when driving a connected car by supplementing the limitation that immediate response is complicated [2,3]. Vehicle to Everything (V2X) is a technology that communicates with vehicles, infrastructure, and people through wireless networks in C-ITS. Connected cars deliver various road and surrounding information using V2X and use the information for autonomous driving along with vehicle sensor data.

Local dynamic map (LDM), a concept devised by the European Telecommunications Standards Institute (ETSI) to store data among various elements constituting C-ITS, stores data such as high-definition map (HD map), weather, and traffic information into four layers [4,5]. LDM can be roughly divided into central LDM and vehicle LDM, and data generated by vehicles and infrastructure are collected from the central LDM. In addition, the central LDM collects and provides additional information necessary for safe driving, such as weather information and accident information, to vehicle LDM. Therefore, in the C-ITS environment, the central LDM acts as a platform to collect data and provide services.

Camera, LiDAR, and GPS sensors mounted on the connected car are critical sensors for providing connected car services and can be used for processing, such as HD map creation and detection of unexpected situations on the road [6–8]. However, in the existing C-ITS environment, the central LDM collects only simple driving information such as speed, position, and direction from the connected car, and the connected car acts as a device to receive services. It is also configured to focus on collecting road situation information from the road side unit (RSU) [9,10]. However, because the RSU is mainly installed on highways, road conditions cannot be determined in areas where the RSU is not installed, such as national highways, and it is time-consuming and expensive to install and use additional equipment for implementing C-ITS. On the other hand, by utilizing the connected car's sensor data while driving, it is possible to reduce the number of additional measuring devices to be installed in the RSU and to build an efficient C-ITS infrastructure. In addition, data collected directly from multiple vehicles on the road improves road situation awareness and the accuracy of traffic information.

The central LDM, which collects vehicle data, has no analytics and processing capabilities implemented. Therefore, we intend to build a platform that provides connected car data by collecting and analyzing all big data of vehicles, such as sensor data and simple driving information collected by the existing central LDM. The platform created by separating the vehicle data collection function of the central LDM can be asynchronously interlocked with the central LDM to provide the processed data according to the central LDM standard to be delivered to the vehicle.

In this paper, we propose a platform, as shown in Figure 1, that includes a big data processing system that collects and processes vehicle big data generated from connected cars and a messaging system that delivers large-capacity vehicle sensor data and traffic information in real time. The proposed system was built as a testbed, and its usefulness and scalability were verified through scenarios. The information analyzed by the proposed vehicle big data analysis system is delivered to the central LDM and vehicle LDM in near real-time to increase the safety of autonomous driving. The proposed big data processing system was built using the Hadoop ecosystem. Vehicle data are stored in the Hadoop distributed file system (HDFS) and MariaDB, and Apache Spark is used for big data analysis. MariaDB's structured data are periodically transferred to and loaded into the HDFS, and the big data and analysis results stored in MariaDB and HDFS are used for the development of various services and business models. Data transfer between the vehicle and the server is realized through the Apache Kafka messaging system. Data transmitted in real-time from the vehicle to the backend server through Kafka is analyzed in real-time using the Spark Streaming function.

The remainder of this paper is structured as follows. Section 2 introduces research trends, e.g., LDM and big data platforms. Section 3 proposes the structure of the entire platform, and Section 4 shows an example of implementing a platform and using autonomous vehicle data. Section 5 concludes the paper.

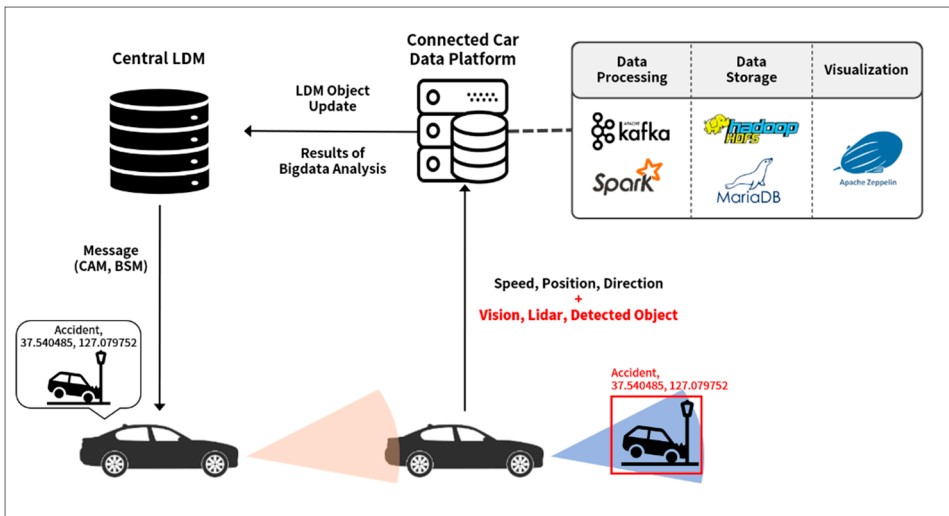

**Figure 1.** Sensor big data processing system for autonomous vehicles in the cooperative intelligent transport system environment.

## 2. Background and Related Works

### 2.1. LDM

The ITS station is an object that affects road traffic and includes vehicles, infrastructure, and mobile devices. LDM is used for rapidly and accurately processing data exchanged between ITS stations in the C-ITS environment, and a concept standard was defined in ETSI. LDM consists of four layers, as shown in Figure 2, and defines HD maps, dynamic information, such as road conditions, accidents, and weather, traffic signals, or moving objects as LDM objects. Static information is stored in Layers 1 and 2, and dynamic information is stored and managed in Layers 3 and 4. The ITS Station implements the C-ITS environment using LDM objects and updates the LDM by sending and receiving objects between stations. Communication message standards such as cooperative acknowledgment message (CAM), event-driven message (DEMN), and road side alarm (RSA) are used when sending and receiving data.

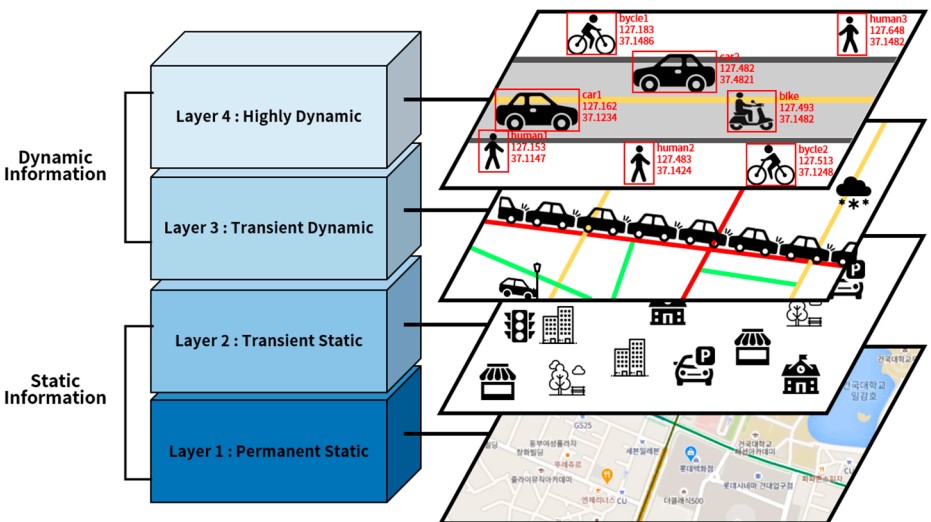

**Figure 2.** Concept of LDM.

The LDM concept for autonomous driving can be broadly divided into vehicle LDM and central LDM. Vehicle LDM is an implementation of the LDM concept in the vehicle. The vehicle LDM, object storage, and control unit for vehicle control are connected through internal communication. In order to

provide accurate information for autonomous cooperative driving, the central LDM receives real-time information from external affiliated organizations in addition to the existing ITS station. External linkage organizations provide weather information, road condition information, and traffic information.

### 2.2. Hadoop Ecosystem

#### 2.2.1. Data Collection Platform

Hadoop is a big data distributed processing framework composed of MapReduce [11], a distributed programming model for processing large amounts of data, and Hadoop distributed file system (HDFS), a distributed data storage system. Hadoop is designated as the core project, and is called Hadoop ecosystem, including various subprojects constituting the framework of Hadoop. As the need for a big data platform increases, the development of Hadoop Ecosystem subprojects is actively progressing. Representative subprojects include Spark, Zookeeper, Kafka, Zeppelin, and Hbase.

#### 2.2.2. Data Processing Platform

Spark is a distributed in-memory processing engine that distributes large amounts of data [12]. Similar to MapReduce, but unlike disk-based MapReduce, memory-based processing solves the problem of slow processing speed. In addition, Spark can process data from various data sources such as DB, CSV, and JSON, and supports streaming functions. The primary processing unit for data analysis inside Spark is a data frame.

Zeppelin is a web-based notebook that supports the data analysis work environment through Spark [13]. In addition, the analysis results can be visualized in a graph and immediately checked.

#### 2.2.3. Data Messaging Platform

Kafka is a subscription-type message distribution system [14]. As shown in Figure 3, it is composed of producer, consumer, and topic. Producers produce messages, and consumers consume messages. In the Kafka system, messages delivered from producers to consumers pass through Kafka broker's topic.

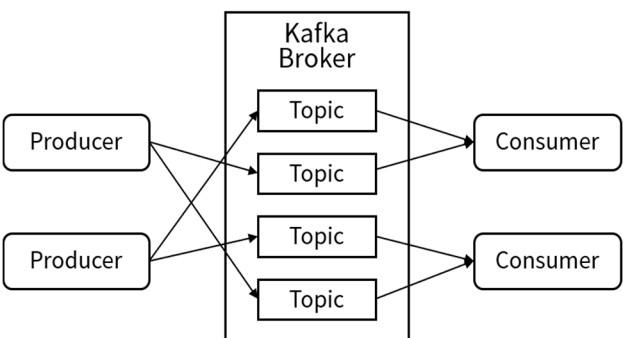

**Figure 3.** Kafka system architecture.

Zookeeper is a distributed process coordination service. It manages distributed processing clusters such as Kafka clusters.

### 2.3. Related Works

Kumar (2018) uses the Hadoop ecosystem to propose a whole process for recognizing, collecting, storing, and analyzing data collected from vehicles as big data [15]. The proposed model consists of collecting all vehicle information through Flume, transmitting it to the HDFS through Kafka, and analyzing and processing it using Spark. This paper only proposes a model in which the Hadoop ecosystem can be applied to the connected car environment, and has the disadvantage of complexity of further use as all messages collected from the vehicles are stored in the HDFS without classification.

Sung et al. (2017) intended to develop a driving environment prediction platform based on road traffic information collected for driver safety. The method is changed to collecting data in real-time from mobile-type sensors rather than from existing fixed-type sensors, thus allowing for various-big-data collection. The data are analyzed on a driving environment analysis platform based on the Hadoop ecosystem, and information is provided through a web environment. However, in the research, vehicle sensor data are limited to road surface temperature, humidity, and vehicle speed.

Among the research on the structure of a platform that collects connected car data, Han (2018) collects vehicle interior data and passenger's body reaction information through OBD and smart watch. The collection and processing phases were designed using GS1, a global standard for business communication, to establish standards for sharing information collected and stored in a connected environment with the outside world. However, the platform proposed in the paper focuses on collecting vehicle information and has the disadvantage that messages shared among connected units are limited to simple vehicle status information such as driving and stopping times.

Islam et al. (2020) aimed to implement a platform for developing connected car services, where the platform users could access data collected from connected cars [16]. Applications developed on the platform directly deliver messages such as BSM to a connected car. Connected car data collected by the platform are message-based information such as vehicle speed, location, and time.

However, in the existing research, autonomous driving environment data are defined as CAN data, speed, and location information collected from the in-vehicle network. A majority of the data collected from a vehicle only serves to deliver the vehicle driving information rather than to derive new information based on the data. Therefore, there is a need for an additional method to utilize vehicle data, and for this, it is necessary to collect various vehicle data such as images and LiDAR data.

Biral et al. (2019) built a C-ITS application for safety using LDM [17]. The C-ITS application proposed in the paper seeks to collect vehicle driving information without equipment by utilizing infrastructure such as ORU installed on the road.

When developing LDM, Roh (2020) points out the limitations of the evaluation method focusing on rapid information delivery rather than information reliability [18]. He proposes a new evaluation method to overcome these limitations.

In the case of existing studies on LDM, the focus is on developing and evaluating the LDM performance. However, research on the process and application of generating information to be provided by LDM is insufficient. Therefore, it is necessary to propose a platform that can apply an algorithm based on various connected car big data and their usage.

Yao et al. (2019) used black box images to detect traffic accidents [19]. It has been shown that traffic accidents or abnormal situations can be grasped from a dynamic camera, rather than a fixed camera image such as CCTV. Therefore, road accidents can be identified with vision data of autonomous vehicles, and vision data should be collected and used on a platform.

## 3. Autonomous Driving Sensor Data-Based C-ITS Environment Architecture

### 3.1. C-ITS Environment

The goal of the proposed C-ITS environment is to incorporate the utilization of vehicle big data. By transmitting vehicle big data from the platform to the central LDM, the accuracy and quality of information can be improved.

Figure 4a is a C-ITS environment based on the existing central LDM that processes the collected data and transmits them to the vehicle LDM. The central LDM detects unexpected situations on the road through the RSU and receives information on the road conditions from the traffic information central. However, the fixed RSU can only gather information on a limited area of about 1 km, making it difficult to assess the road conditions accurately. Another disadvantage is that additional equipment is required to utilize the RSU for C-ITS. However, vehicles are also equipped with sensors similar to the

RSU, but unlike the RSU, only simple data such as vehicle speed and direction angle are transmitted to the central LDM.

Figure 4b describes the C-ITS environment that compensates for the disadvantages of the existing C-ITS environment shown in Figure 4a with sensor data from an autonomous vehicle, a moving IoT equipment. The data collected from the vehicle are transmitted to the central LDM after recognizing a broader scope than the infrastructure. In addition, when the big data platform recognizes information on weather and traffic situation of the area where the vehicle is driving in real time, it transmits the data to an external affiliated organization to supplement the real-time information.

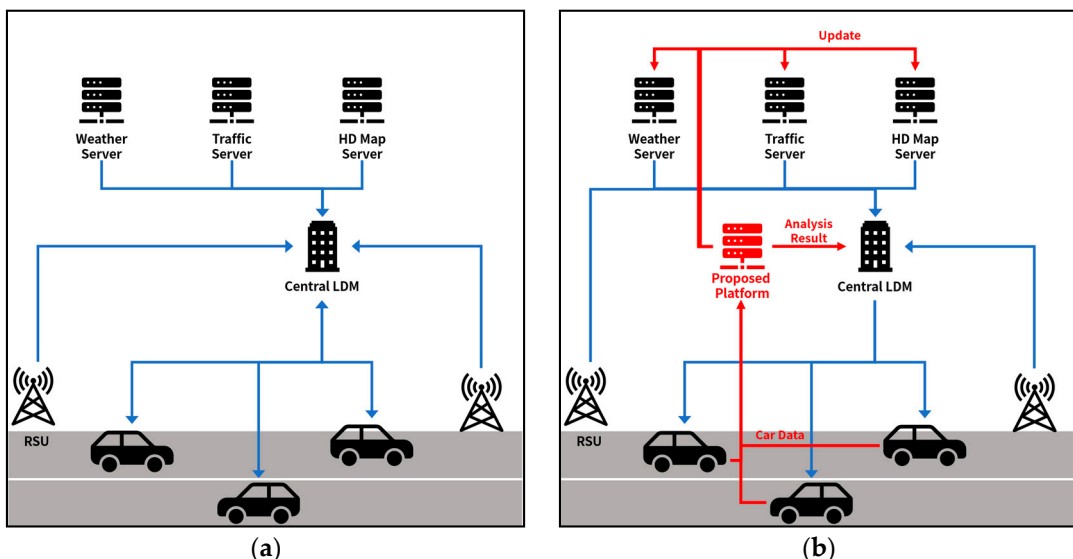

**Figure 4.** (**a**) Existing C-ITS environment diagram; (**b**) C-ITS environment with the proposed platform diagram.

The internal structure of the big data-based LDM linkage platform for autonomous cooperative driving is composed mainly of three systems, as shown in Figure 5. The vehicle system collects data from the equipment installed in the vehicle and vehicle driving data and drives autonomously through real-time processing. It also communicates with surrounding vehicles and the infrastructure in the vehicle system to provide the basis for cooperative driving. The platform is the core system of the proposed C-ITS environment, consisting of several modules to store, process, and analyze big data collected from the vehicles. Analysis results are transmitted to the vehicles and users again using a messaging system or are converted into a structure suitable for utilization and provided to affiliated organizations outside C-ITS. The service system consists of a messaging system that provides analysis results to the central LDM and a search module that allows users to access the server and check data directly. The search module provides a function to check and download data suitable for various research purposes to utilize the collected data.

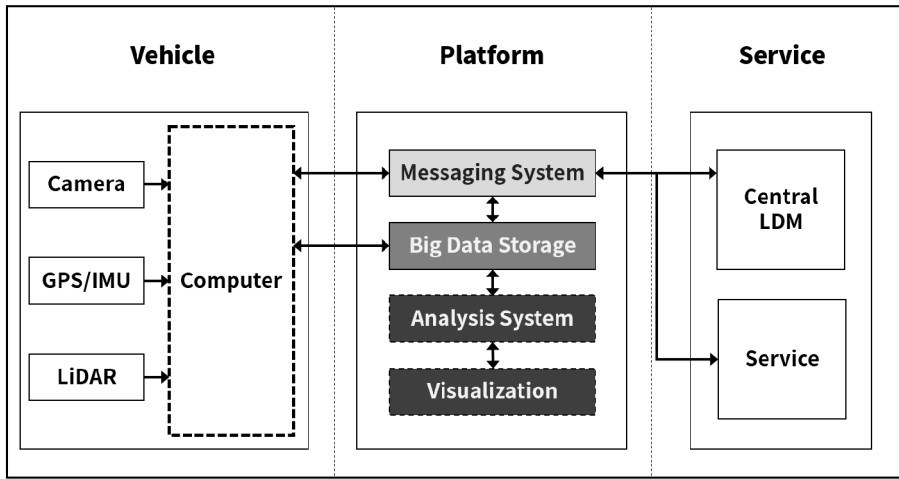

**Figure 5.** Overview of the proposed C-ITS environment.

### 3.2. Vehicle System in the Proposed C-ITS Environment

The vehicle system consists of equipment for autonomous driving, the process of detecting and processing sensor data, controlling the vehicle, and communicating with the outside. Figure 6 describes the vehicle system within the system.

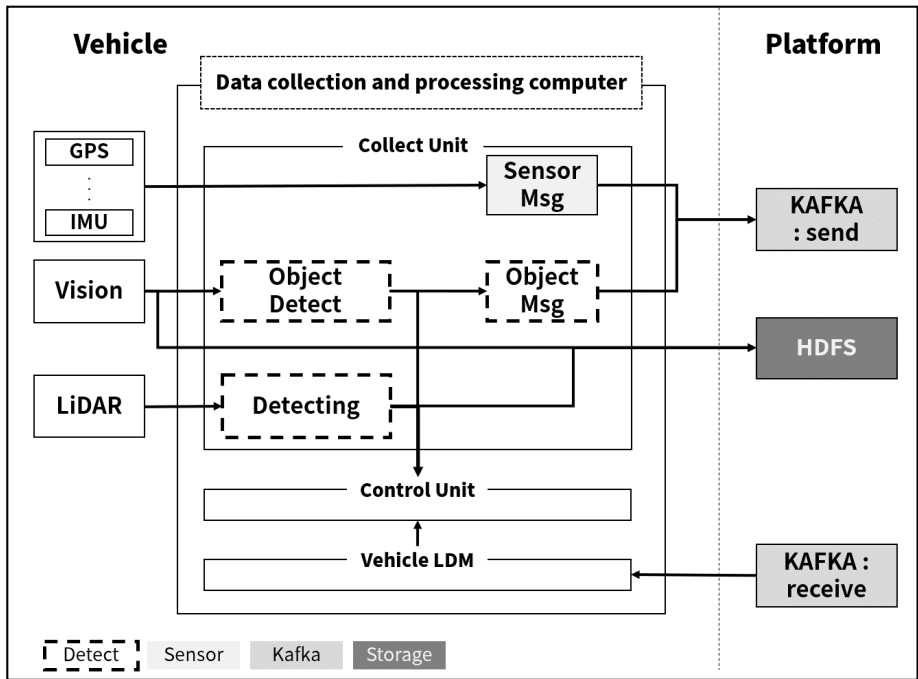

**Figure 6.** Architecture of the vehicle system in the proposed C-ITS environment.

Equipment essential for autonomous driving includes sensors, Vision, and LiDAR. Sensors include GPS and IMU for vehicle positioning, and Vision mainly uses cameras as equipment. The sensor data generated by each device are organized in a different data format, so the data processing method should be different for each device.

The process module mainly consists of a computer, control unit, and vehicle LDM. The computer converts the data collected from the sensor into JSON, a lightweight message format suitable for transmission to the Kafka server, and sends them to the Kafka server. At the same time, it transmits the vehicle status information and the driving information. The data collected by Vision are transmitted to the control unit after object identification for autonomous driving. Simultaneously, the identified

object information is converted into JSON and sent to the Kafka server. The raw images collected in Vision are delivered to Kafka servers in JSON format while driving. After driving is finished, they are transmitted to the HDFS, a large-capacity data storage in the form of video files. LiDAR data are also transferred directly to the HDFS, similar to Vision's raw images.

### 3.3. Platform in the Proposed C-ITS Environment

The platform configured by using the Hadoop ecosystem is described in Figure 7. The platform can be primarily divided into a data collection module, data storage module, and data processing and analysis module.

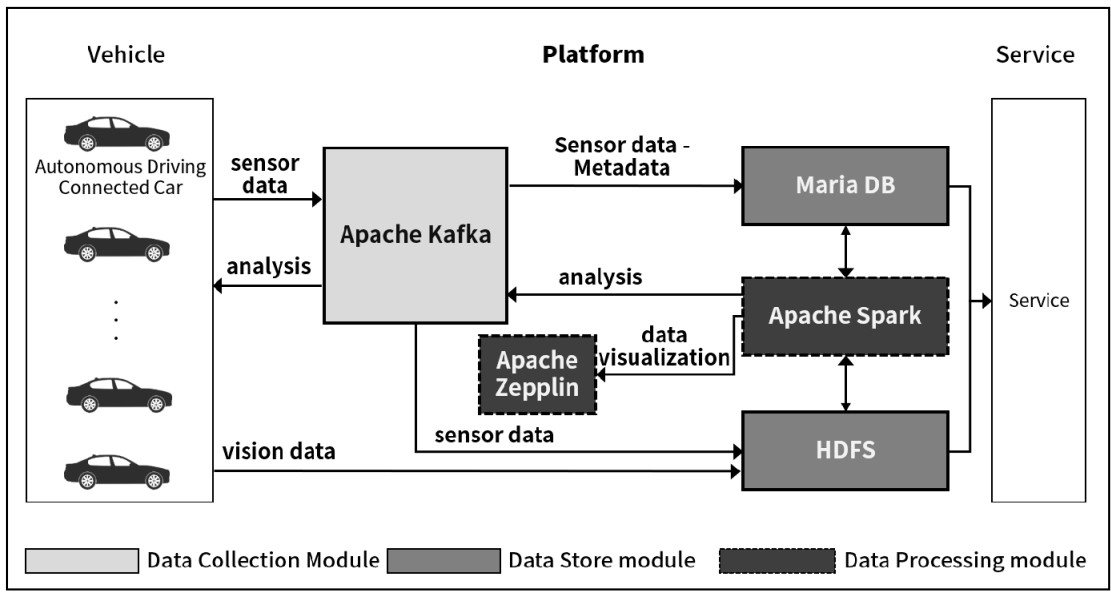

**Figure 7.** Architecture of the platform in the proposed C-ITS environment.

### 3.3.1. Data Collection Module

Kafka is distributed across multiple servers, and the risk of message loss is low because messages are replicated and stored on multiple servers even during normal times. Therefore, the Kafka messaging system is the core system of the data collection module. In the data collection module, an autonomous vehicle driving on the road functions as a Kafka publisher and delivers the data to the Kafka server. In this process, the data collected from each sensor are collected from the topic responsible for each sensor. If the topic for each sensor is separated, the transmitted data can be parsed appropriately and used for analysis. Even if a problem occurs in one topic, other data can be collected continuously. In addition, Kafka is highly compatible and can not only connect within the platform, e.g., MariaDB, HDFS, and Spark, but also deliver messages to the service system outside the platform. Therefore, the service system serves as Kafka's consumer. In addition, as multiple consumers can access a topic, the central LDM, vehicles, and external affiliated organizations can simultaneously access the analysis topic.

### 3.3.2. Data Store Module

The data storage module consists of Maria DB and HDFS, the representative loading framework of the Hadoop ecosystem. It is appropriate to store the structured data collected from vehicles in a database that can be systematically managed after designating a schema. Among several databases, MariaDB is used in the proposed system because it is a lightweight, fast, and large relational database management system (RDBMS) that guarantees high stability. Conversely, the HDFS is more suitable for storing large amounts of data, raw image data, or Vision's LiDAR data collected from vehicles. In

addition, owing to the rapidly increasing characteristics of vehicle data, data collected from MariaDB within a certain period are periodically transferred to the HDFS.

Information such as GPS and IMU data, which are sensor data, vehicle driving information, object type identified by Vision, and weather, are defined in the schema, as shown in Figure 8.

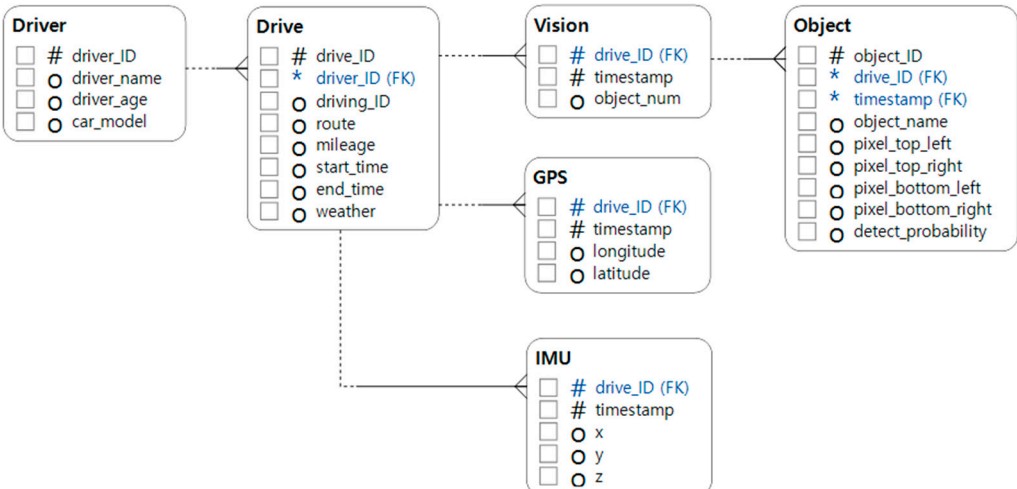

**Figure 8.** Schema of the RDBMS.

As the data of each table of MariaDB are continuously generated while driving, it is designed in a 1:M structure. The Driver table contains the driver's identification information, and the Drive table includes information on the driver's driving situation. Therefore, the Drive table contains all data from the moment the engine is turned on to the moment the engine is turned off. The GPS and IMU tables store each sensor data value generated during driving, and the Vision table records the timestamp when an object is detected while driving. Object information detected by camera is recorded in the Object table based on the timestamp of the Vision table. The position of the object in the frame is expressed as the bounding box's pixel position, and based on this, the size of the object and its coordinates in the real coordinate plane can be known.

The HDFS stores Vision raw image data or LiDAR delivered through Kafka. In addition, raw image data of Vision can be converted into an image file after reprocessing and stored, and preserved data transferred from Maria DB are also loaded.

### 3.3.3. Data Processing Module

The processing and analysis module visually grasps the collection status of vehicle data, processes data requiring real-time processing through streaming, and is used when analyzing large-scale data for specific purposes. Therefore, Spark and Zeppelin are used for processing and analysis modules. Spark is an open-source cluster computing framework that allows users to analyze data results in multiple formats. At this time, data storage formats such as CSV and parquet are also used, but because data stored in the HDFS can be analyzed directly as a source, it shows fast throughput. In addition, Spark provides a streaming function. This function is used when analyzing data collected in real time through Kafka. For example, GPS data collected from vehicles can be streamed, and the real-time location and speed of each vehicle can be identified to determine whether a specific road is congested. When real-time road conditions are identified through Spark, this information is transmitted to the central LDM, which can be combined with existing information and obtain high accuracy by analyzing the road conditions from various angles based on combined information. In addition, it is possible to derive analysis results by reading vehicle data for a specific period from the HDFS and MariaDB.

This process can be performed directly from Spark or by accessing Spark through Zeppelin. In addition, Zeppelin can be used when it is necessary to output a visualized analysis result.

*3.4. Service System in the Proposed C-ITS Environment*

The service system refers to ITS stations other than vehicles that deliver LDM objects to vehicles. Therefore, the service system is connected using a messaging system because it includes the central LDM and an external linkage authority.

In addition, users must be able to access the collected data for the system utilization. The utilization refers to downloading data and analyzing data for transmission to the central LDM. In addition, the service system can be connected to Java applications, web servers, and maps according to the function to be implemented.

## 4. Results

This research builds a testbed using a server and a test vehicle, as shown in Figure 9, to implement the proposed C-ITS environment. Tables 1 and 2 describe the testbed.

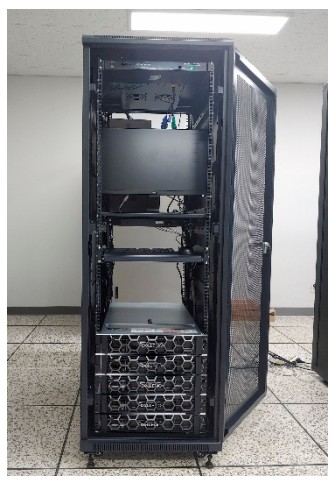 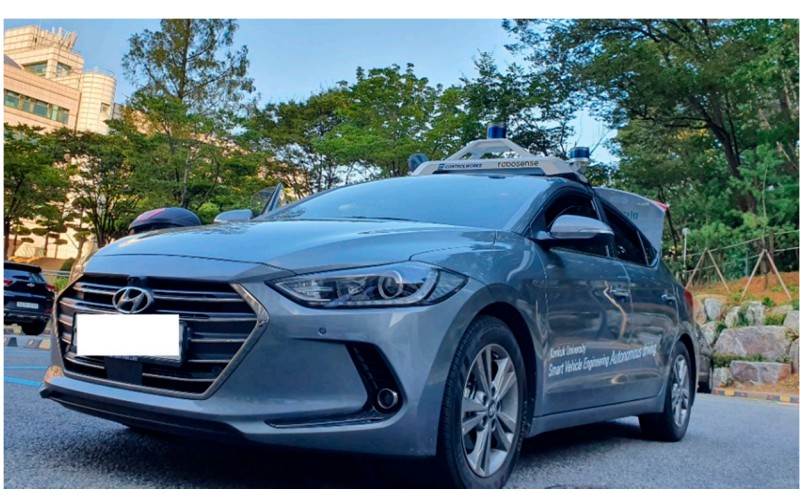

(**a**)          (**b**)

**Figure 9.** (**a**) Server; (**b**) test vehicle.

**Table 1.** Server specifications.

| Type | Model |
| --- | --- |
| Server | PowerEdge R740 Server |
| Processor | Intel Xeon Silver 4208 2.1 G |
| RAM | 16 GB |
| Core | 8 C |
| HDD | 1.8 TB |
| EA | 5 |

**Table 2.** Test vehicle and sensor specifications.

| Type | Model |
| --- | --- |
| Test Vehicle | Hyundae Avante |
| GPS and IMU | EVK-M8U |
| Camera | oCam_5CRO-U |

### 4.1. Implemented Platform and C-ITS Environment

Figure 10 shows the structure diagram of the actual implemented C-ITS environment and platform, and Table 3 shows the SW and OS information used in the platform configuration.

The proposed autonomous vehicle sensor big data-based C-ITS environment configures the testbed using Docker, OS-level virtualization. Using Docker, we created an environment similar to multiple server nodes by creating numerous containers, such as processes running in an isolated space. The vehicle system uses three ROS nodes to implement a connected car environment [20]. Each ROS node acts as a vehicle, and various connected car environments can be configured by increasing or decreasing the number of the ROS nodes.

The platform was configured using multiple nodes to process the big data stably. The Hadoop cluster consists of a DataNode that stores data and a NameNode that stores where they are stored. In order to improve stability, a secondary NameNode that duplicates the data of the NameNode was also implemented. When the problem of data storage space shortage occurs owing to an increase in the number of vehicles in the future, it can be solved by adding a DataNode. The cluster is also configured to expand by adding Kafka nodes as the number of vehicles increases.

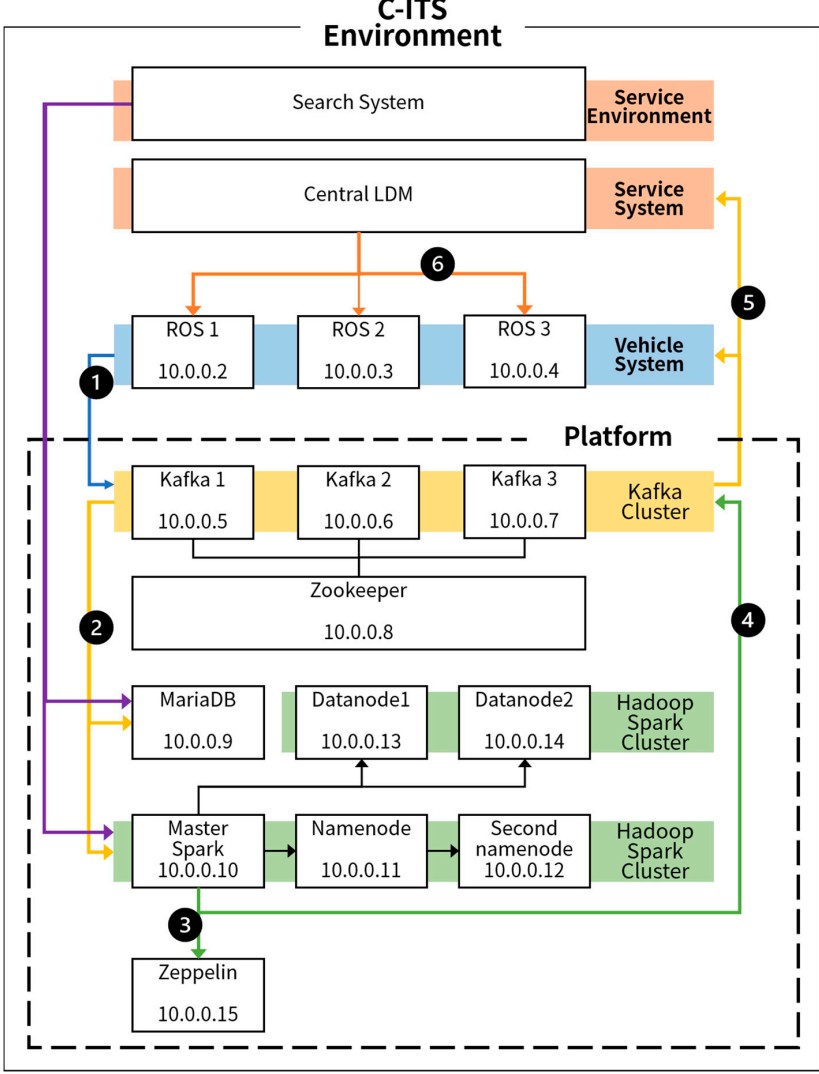

**Figure 10.** Implementation of the proposed C-ITS environment.

**Table 3.** Software-platform information.

| OS/Software | Version |
|---|---|
| CentOS | 7 |
| ROS | melodic |
| Docker | 1.13.1 |
| Kafka | 2.13 |
| Spark | 2.3.3 |
| YOLO | v2 |

*4.2. Implementation—Collection and Transmission of Driving Data*

To implement 1 in Figure 10, an autonomous vehicle owned by the department drove on campus, and ROS collected the sensor data. The vehicle is equipped with sensors for autonomous driving, and the sensors are collected as ROS to control the car. In order to realize the actual autonomous driving environment, all driving records are stored in ROSBAG, the record file of ROS, and implemented similarly to the connected car environment using multiple ROSBAGs.

The replayed ROSBAG outputs the stored sensor data and delivers them to the Kafka server through the ROS-Kafka connector connected to the ROS topic. There are three main types of data collected from vehicles in the system: image, IMU, and GPS data. Therefore, Kafka topics are created according to the data from the ROS topics.

Autonomous vehicles identify objects through image recognition while driving and output the results or pass them over to the control unit. 2 in Figure 10 proceeds with object detection using Tensorflow and YOLO based on the information received from the ROS-Kafka topic in 1. The detection results and sensor data collected from ROS are converted to JSON format with a schema for DB storage and passed through Kafka topics linked to each table in MariaDB. In addition, the video data are converted into a video file after collecting and detecting real-time raw image files and stored in the HDFS.

Figure 11 describes the elaborated ROS topic and the Kafka topic structure for collecting and transmitting driving data, and an example of the exchanged message is shown in Figure 12.

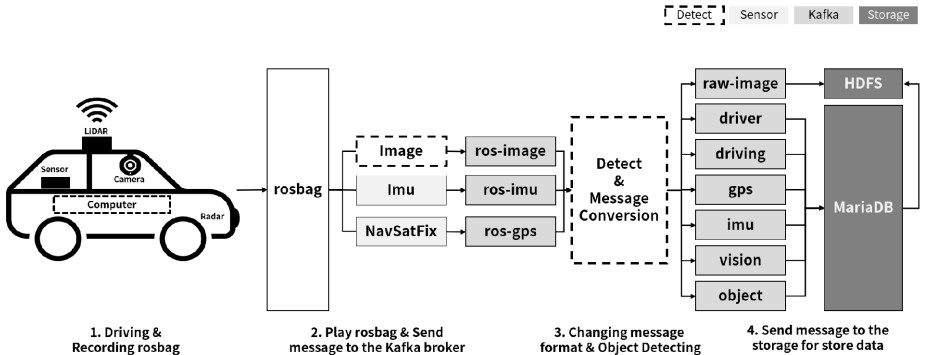

**Figure 11.** Autonomous vehicle data collection process.

{"schema":{"type":"struct","fields":[{"type":"string","optional" : true, "field" : "drive_ID"}, {"type" : "string", "optional" : true, "field" : "timestamp"}, {"type" : "float", "optional" : true,"field" : "longitude"}, {"type" : "float","optional" : true, "field" : "latitude"}],"optional" : "false","name":"gps"},"payload":{"drive_ID": "idpC_2", "timestamp": "2020-09-05 12:26:06.875774", "latitude": 37.5205022, "longitude": 126.6105801}} ❶
{"schema":{"type":"struct","fields":[{"type":"string","optional" : true, "field" : "drive_ID"}, {"type" : "string", "optional" : true, "field" : "timestamp"}, {"type" : "float", "optional" : true,"field" : "longitude"}, {"type" : "float","optional" : true, "field" : "latitude"}],"optional" : "false","name":"gps"},"payload":{"drive_ID": "idpA_1", "timestamp": "2020-09-05 12:25:57.873334", "latitude": 37.5205001, "longitude": 126.6105771}} ❷
{"schema":{"type":"struct","fields":[{"type":"string","optional" : true, "field" : "drive_ID"}, {"type" : "string", "optional" : true, "field" : "timestamp"}, {"type" : "float", "optional" : true,"field" : "longitude"}, {"type" : "float","optional" : true, "field" : "latitude"}],"optional" : "false","name":"gps"},"payload":{"drive_ID": "idpB_3", "timestamp": "2020-09-05 12:26:10.772530", "latitude": 37.5205018, "longitude": 126.6105785}} ❸

**Figure 12.** Message from ROS to the database.

### 4.3. Implementation—Data Analysis and Visualization

Spark analyzes the data based on the stored data and visualizes them using Zeppelin, as shown in Process 3 in Figure 10.

Spark creates Dataframe based on the data accumulated in Maria DB and HDFS. Subsequently, writing the code in Zeppelin's notebook, which is designed to facilitate this process, creates a temporary view 'tmpTable' based on the data frame for an SQL query. Finally, users can visually understand the types of items detected while driving based on the temporary view of the object table of MariaDB, as shown in Figure 13.

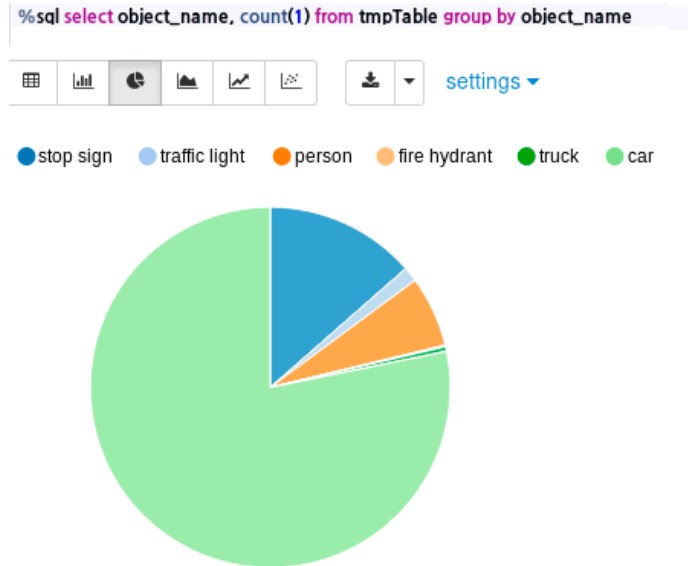

**Figure 13.** Zeppelin-based data visualization.

### 4.4. Implementation—Central LDM Linkage Process

Thereafter, 3 and 4 processes proceed, and the analysis result is delivered to the Kafka cluster. Kafka clusters organize topics that can deliver messages to vehicles and service environments outside the server for analysis-result-based services. Therefore, the process of transmitting the information to the vehicle LDM by linking the message received in 5, the information collected by the central LDM from the server, and external information is 6.

### 4.5. Implementation—Latency Evaluation

As shown in Table 4, the latency evaluation of the proposed platform and C-ITS environment was performed. The overall latency evaluation process is the same as the autonomous vehicle data collection process in Figure 11. The process of transferring the sensor data collected from the vehicle to the Python application through Kafka is the same as steps 1–3 in Figure 11, which takes 14 ms on average. Additionally, after converting the message format in a Python application, the data is transferred to the topic connected to the Database. At this time, Central LDM can also receive data simultaneously based on Kafka's ability to subscribe to the same topic. This process also had an average latency of 14 ms.

**Table 4.** Latency evaluation of the proposed C-ITS environment.

| Type of Time | ROS to Python Application | Python Application to Database | Overall Process |
|---|---|---|---|
| Average | 14 ms | 14 ms | 28 ms |
| Maximum | 43 ms | 184 ms | 213 ms |
| Minimum | 5 ms | 4 ms | 14 ms |

The latency maximum situation occurred when data was first transferred from the application to the database. This is the first time that Kafka and the database connector are connected, causing a delay. In addition, when using the Kafka server for both ROS to Python application and Python application to Database processes, the proposed C-ITS environment guarantees stable latency because the maximum latency for occupant safety does not exceed 200 ms, and an average speed of 14 ms is guaranteed [16].

### 4.6. Implementation—Road Surface Condition Recognition Example

On cracked roads, the vehicle vibrates while driving, reducing the passenger's ride comfort and increasing the risk of accidents. Therefore, continuous monitoring and management of cracked roads are required. As the acceleration sensor IMU provides acceleration in the x, y, and z axes, the degree of the vibration of the vehicle can be determined through the sensor value analysis. Therefore, the traffic center can check the road surface condition using the vehicle IMU sensor data. There is an example that recognizes the road surface condition using the built system.

① It delivers the IMU data collected from the vehicle to the IMU topic of Kafka.
② The data in MariaDB from the IMU topics are stored. At the same time, Spark uses the streaming function to find data with significant fluctuations in the y- and z-axis values in real time.
③ When abnormal data are detected, Spark delivers information on the location of the occurrence to Kafka.
④ The service environment collects messages delivered from Kafka-service topics.
⑤ The information collected for road management and monitoring is displayed on a web-based map in real time.

The result of recognizing the road surface condition using the built system is shown in Figure 14. Currently, the driving information and the IMU sensor data are used. However, if an advanced object detection algorithm is applied to the vehicle control, the data in the object table of the RDBMS and the sensor data are fused to obtain the accurate road condition information.

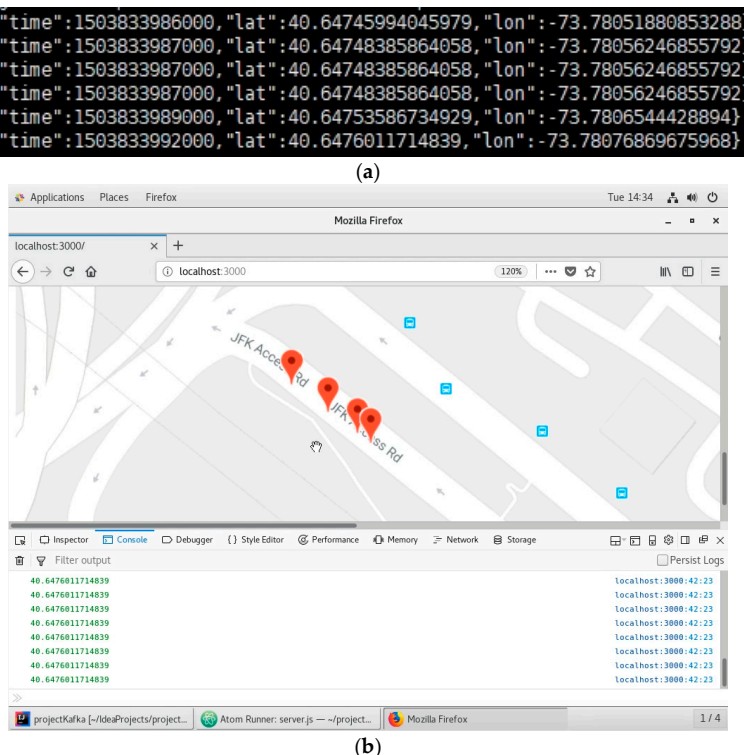

**Figure 14.** (**a**) Location information transmitted when Spark detects abnormal data; (**b**) web-based visualization of the messages delivered to the central LDM.

## 5. Conclusions and Future Work

This study proposes an autonomous driving sensor big data-based C-ITS environment to improve the safety of autonomous vehicles. Unlike the current C-ITS environment, which highly depends on the infrastructure, a structure that utilizes big data for autonomous vehicles and connected cars was proposed and implemented based on the Hadoop ecosystem. The implemented Hadoop ecosystem-based LDM linkage platform collects images, sensors, and LiDAR data from ROS in charge of controlling autonomous vehicles. It delivers them to Kafka, a distributed messaging system that guarantees fast latency. Messages delivered from the vehicle system to the platform through Kafka are stored in MariaDB for structured data and HDFS for unstructured big data. The platform analyzes and processes the loaded data using Spark and visualizes the results with Zeppelin. The data processed in the platform transmit a message to the central LDM through Kafka for vehicle services. As the proposed platform is constructed using the Hadoop ecosystem, which can be distributed, it is possible to expand if the vehicle big data are increased and to ensure stability and fault tolerance.

Additional studies that can be conducted based on this study are as follows.

(1) For a systematic management of the vehicle big data, metadata of MariaDB and HDFS are created and constructed. As the big data accumulate, it is difficult to obtain the location and type of the existing data, so it is crucial to optimize the data management. Therefore, in future work, metadata storage using NoSQL should be investigated.

(2) In order to utilize LDM, a real-time system can be configured, and a process of transmitting and receiving messages conforming to the standard can be performed. It can also be linked with research on security enhancement of the message transmission and reception processes.

**Author Contributions:** Conceptualization, A.Y. and C.M.; methodology, A.Y. and C.M.; software, A.Y., S.S. and J.L.; validation, A.Y., S.S. and J.L.; investigation, A.Y. and S.S.; data curation, A.Y. and S.S.; writing—original draft preparation, A.Y.; writing—review and editing, A.Y., S.S., J.L. and C.M.; visualization, A.Y.; supervision, C.M.; project administration, C.M.; All authors have read and agreed to the published version of the manuscript.

**Funding:** This paper was supported by the Korea Institute for Advancement of Technology (KIAT) grant funded by the Korea Government (MOTIE) (N0002428, The Competency Development Program for Industry Specialist).

**Conflicts of Interest:** The authors declare no conflict of interest.

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
