# Peer review of "Implementation of a Sensor Big Data Processing System for Autonomous Vehicles in the C-ITS Environment"

_applsci, doi:10.3390/app10217858_

Round 1

Reviewer 1 Report

The paper presents a Hadoop based data processing platform for the C-ITS domain. The proposed work is relevant and methodologies standard. The authors have the infrastructure to carry out the work and that must be properly harnessed. However, the few concerns I have that needs to be addressed if the paper is to be published. They are:

  1. In the abstract, not sure what the "however" is for. The motivation is not coming out clearly in the Abstract and similarly in the Introduction, there is no justification of the work advancing knowledge in the context of current state-of-art. 
  2.  Definitions must be more formal . Such as for example in the Abstract, the definition of Big Data is inappropriate. 
  3.  The Related work lacks an intensive literature review and how the presented work is different . For, example there is a number of similar work such as https://journals.sagepub.com/doi/abs/10.1177/0361198120917146  
  4.  The claim on line 65 and 66 regarding 5G needs proper justification and similarly such claims.
  5.  The proposed platform is embedded into the existing infrastructure, hence there is an urgent need for evaluating latency and how that affects any application. Having said that the Evaluation and results section really needs work on to check the system on metrics that can be evaluated against existing work as well as be used a a benchmark for future work. 

The paper has minor English language and grammatical errors that needs checking. 

Reviewer 2 Report

The paper presents an improved architecture for C-ITS that relying on the use of the vehicles' information in order to update the Central LDM information.

In the paper an analysis of the scalability of the system is missing, I suggest the authors add this analysis.

Further minor changes need to be made, in particular:

the paper lacks clarity in the abstract, I suggest to revise the text of the abstract.

"the proposed" -> "The proposed" line 76.

Add one picture at the beginning to understand as soon as possible the architecture.

ETSI acronym missing line 91.

Figure 1 is not cited in the text.

Figure 5 shows the element "process" and "computer", are they indicated properly?

Line 210 and 211 says that raw data are delivered in JSON, but it is not possible.

In Figure 6 "data collection module" and "data store module" are indicated with too much similar color not suitable for b&w printing.

DBMS what is it?

Figure 9 it is not clear what runs in the vehicle and what inside the central LDM. What is the platform?
